# PNLDC1, mouse pre-piRNA Trimmer, is required for meiotic and post-meiotic male germ cell development

Toru Nishimura[1,†], Ippei Nagamori[2,†], Tsunetoshi Nakatani[2], Natsuko Izumi[3], Yukihide Tomari[3,4], Satomi Kuramochi-Miyagawa[2,5,*] iD & Toru Nakano[1,2,5,**] iD

## Abstract

PIWI-interacting RNAs (piRNAs) are germ cell-specific small RNAs essential for retrotransposon gene silencing and male germ cell development. In piRNA biogenesis, the endonuclease MitoPLD/Zucchini cleaves long, single-stranded RNAs to generate 5′ termini of precursor piRNAs (pre-piRNAs) that are consecutively loaded into PIWI-family proteins. Subsequently, these pre-piRNAs are trimmed at their 3′-end by an exonuclease called Trimmer. Recently, poly(A)-specific ribonuclease-like domain-containing 1 (PNLDC1) was identified as the pre-piRNA Trimmer in silkworms. However, the function of PNLDC1 in other species remains unknown. Here, we generate *Pnldc1* mutant mice and analyze small RNAs in their testes. Our results demonstrate that mouse PNLDC1 functions in the trimming of both embryonic and post-natal pre-piRNAs. In addition, piRNA trimming defects in embryonic and post-natal testes cause impaired DNA methylation and reduced MIWI expression, respectively. Phenotypically, both meiotic and post-meiotic arrests are evident in the same individual *Pnldc1* mutant mouse. The former and latter phenotypes are similar to those of MILI and MIWI mutant mice, respectively. Thus, PNLDC1-mediated piRNA trimming is indispensable for the function of piRNAs throughout mouse spermatogenesis.

**Keywords** piRNA; PIWI; PNLDC1; spermatogenesis; Trimmer
**Subject Categories** Development & Differentiation; RNA Biology

See also: **AW Bronkhorst & RF Ketting** (March 2018)

## Introduction

PIWI-interacting RNAs (piRNAs), germ cell line-specific non-coding small RNAs comprising 24–31 nucleotides (nt), are essential for germ cell development and male fertility in mice [1,2]. piRNAs bind to PIWI subfamily proteins and play essential roles in transposon silencing at the transcriptional and post-transcriptional levels [3,4]. In mice, there are three PIWI subfamily proteins, which differ in their timing of expression and biological function: MIWI (mouse PIWI), MILI (MIWI like), and MIWI2 [5]. MILI is constitutively expressed in embryonic testes from the primordial germ cell to round spermatid stage except in leptotene and zygotene stages of meiotic prophase [6,7]. Expression of MIWI2 begins around the same time as that of MILI but ceases at the spermatogonia stage just after birth [8,9]. It is notable that both MILI and MIWI2 are essential for piRNA production and subsequent DNA methylation of retrotransposons in embryonic male germ cells [9–12]. MIWI is expressed during the later stages of spermatogenesis, that is, only from the pachytene spermatocyte to spermatid stage, in post-natal testes [7,13]. Mutation of these genes resulted in infertility via apoptosis of male germ cells [12–14]. Cell death in MILI- and MIWI2-null mice occurred at meiotic division before pachytene spermatocyte development. In contrast, MIWI-null mice show cell death at the post-meiotic round spermatid stage. In addition, there is a clear difference between the origins of embryonic piRNAs and pachytene piRNAs; namely, the majority of MILI- and MIWI2-bound piRNAs in embryonic testes are derived from retrotransposon genes, whereas MIWI-bound piRNAs in post-natal testes are derived from intergenic regions [9,11,15–17].

In embryonic male germ cells, piRNA biogenesis consists of primary and secondary processing [9,11]. The initial step in primary piRNA biogenesis is thought to be the cleavage of long transcripts derived from retrotransposon loci by the endonuclease mitochondrial phospholipase D (MitoPLD)/Zucchini, which produces piRNA

1   Graduate School of Frontier Biosciences, Osaka University, Suita, Osaka, Japan
2   Department of Pathology, Osaka University, Suita, Osaka, Japan
3   Institute of Molecular and Cellular Biosciences, The University of Tokyo, Bunkyo-ku, Tokyo, Japan
4   Department of Computational Biology and Medical Sciences, Graduate School of Frontier Sciences, The University of Tokyo, Bunkyo-ku, Tokyo, Japan
5   CREST, Japan Science and Technology Agency (JST), Saitama, Japan
    *Corresponding author. Tel: +81 6 6879 3722; E-mail: smiya@patho.med.osaka-u.ac.jp
    **Corresponding author. Tel: +81 6 6879 3720; E-mail: tnakano@patho.med.osaka-u.ac.jp
    †These authors contributed equally to this work

intermediates with defined 5′ ends [18,19]. The piRNA intermediates are loaded onto MILI proteins and cleaved again by MitoPLD/Zucchini, producing precursor piRNAs (pre-piRNAs), with lengths of ~30–40 nt [20–22]. The 3′ ends of pre-piRNAs are then trimmed in an exonucleolytic manner, resulting in the mature piRNA length, and are finally 2′-O-methylated [23,24]. A characteristic signature of primary piRNAs is the strong bias for a uridine at the first nt position (1U) [3].

In contrast, secondary piRNA biogenesis commences by the recognition of complementary transcripts by MILI-bound 1U primary piRNAs [11]. The slicer activity of MILI cleaves the complementary strand at the position complementary to the $10^{th}$ nt of the guide piRNA, triggering the production of antisense secondary piRNAs, which are loaded onto either MILI or MIWI2. Because of this slicer-mediated biogenesis mechanism, primary and secondary piRNAs bear precise 10-nt overlaps at their 5′ ends [11,25]. Moreover, antisense secondary piRNAs show bias for an adenine at the $10^{th}$ nt position (10A) from the 5′ end. Like primary piRNAs, secondary pre-piRNAs are also trimmed and 2′-O-methylated at their 3′ ends after loading onto PIWI proteins [23,26,27].

piRNA biogenesis in post-natal male germ cells differs strikingly from that in embryonic cells, because the majority of piRNAs are produced only by primary biogenesis after birth [28,29]. The post-natal piRNAs can be divided into pre-pachytene and pachytene piRNAs based on their timing of expression and their corresponding locus in the genome [28]. Pachytene piRNAs are loaded mainly into MIWI but also into MILI [28,29], and unlike embryonic piRNAs, pachytene piRNAs have a strong 1U but no 10A bias, reflecting their primary biogenesis-dependent mechanism [28,29].

A previous biochemical study proposed that a putative 3′–5′ exonuclease, Trimmer, mediates 3′-end trimming of pre-piRNAs into mature piRNAs [24]. Recently, PNLDC1 was identified as the Trimmer protein in silkworms [23]. The trimming reaction by Trimmer requires the Tudor domain protein BmPapi (TDRKH or TDRD2 in mammals). Mice lacking *Tdrkh* show meiotic arrest, with reduced levels of mature piRNAs and accumulation of longer piRNAs [22], indicating that 3′-end trimming plays a critical role in the function of mouse embryonic piRNAs. In *Caenorhabditis elegans*, a canonical poly(A)-specific ribonuclease (PARN), PARN1, trims the 3′-ends of worm-specific 21U-RNAs [30]. In mice, PARN is expressed ubiquitously; however, the expression of mouse *Pnldc1* is restricted to testes [31], suggesting PNLDC1 as a candidate pre-piRNA trimming enzyme in mice. In this study, we generated *Pnldc1* mutant mice and analyzed the function of PNLDC1. Notably, male germ cells in the mutant mouse lines showed two types of abnormalities, at the meiotic and post-meiotic stages, in the same individual. These abnormalities can be attributed to the trimming deficiency in both embryonic and post-natal piRNA production.

# Results and Discussion

## Two types of abnormalities in *Pnldc1* mutant testes

*Pnldc1* mutant mice were produced using the CRISPR/Cas9 system. Injection of a single-guide RNA targeting *Pnldc1* exon 3 produced

*Pnldc1*^mt/mt^ mice harboring an 11-bp deletion in *Pnldc1* (Fig 1A). The mice were viable, and their body weights were comparable with those of the control mice (Fig EV1A). The size of the testes of 8-week-old *Pnldc1*^mt/mt^ mice was approximately half that of their control siblings (Fig 1B). Arrest of spermatogenesis at both the meiotic and post-meiotic phases in the same individual was observed (Fig 1C and D). The frequencies of the meiotic and post-meiotic phenotypes were 49% (228/461) and 51% (233/461), respectively, and no sperm was observed in the epididymis (Fig 1C). This phenotype suggested that spermatogenesis was reduced to approximately 50% around the pachytene spermatocyte stage and was blocked completely at the post-meiotic phase. The meiotic and post-meiotic phenotypes of the *Pnldc1*^mt/mt^ mice were quite similar to those of MILI- and MIWI-null testes, respectively.

We generated another *Pnldc1* mutant mouse line to confirm the biological function of PNLDC1 (Fig EV1B). Mutant mice containing an ~800-bp insertion in exon 7 of *Pnldc1* (Fig EV1B–D) exhibited smaller testes and no sperm (Fig EV1E and F). Two types of spermatogenic defects were observed, as in the *Pnldc1*^mt/mt^ mice, and the frequencies of the meiotic and post-meiotic phenotypes were 47% (232/490) and 53% (258/490), respectively. These two mutant lines exhibited essentially identical phenotypes, and hereafter, *Pnldc1*^mt/mt^ mice were used as the representative line.

## Mouse PNLDC1 as the pre-piRNA Trimmer protein of embryonic piRNAs

Small RNAs were obtained from embryonic day 16.5 (E16.5) testes and subjected to deep sequencing analysis. The distribution of small RNAs with lengths of 24–50 nt was categorized as indicated in Fig EV2A [32]. Although there were no significant differences in the numbers of small RNAs corresponding to exons, introns, 5′ UTRs, and 3′ UTRs, mutant mice showed a significant decrease in the number of small RNAs corresponding to repetitive sequences, such as long terminal repeats (LTRs) and long interspersed elements (LINEs) (Fig EV2A).

Next, the size distribution of small RNAs with repetitive sequences derived from retrotransposons was analyzed (Fig 2A). In the control embryonic male germ cells, the majority of the small RNAs derived from repetitive sequences were 24–31 nt in length. In striking contrast, in the mutant embryonic testes, the small RNA sizes were broadly distributed from 24 to 50 nt. Analysis of the first nt of the small RNAs from mutant mice showed a strong 1U bias, as did those from control mice (Fig 2B). These data strongly suggest that these small RNAs in the mutant mouse embryos are untrimmed pre-piRNAs.

Because the major function of embryonic piRNAs is DNA methylation of intracisternal A particle (IAP)-1Δ1 and LINE-1 retrotransposons [9,11,12,33,34], we next analyzed the piRNAs to IAP and LINE-1 types A and Gf. While small RNAs that mapped to IAP and LINE-1 transposons were mildly decreased, the proportions of 1U sense and 10A antisense small RNAs were not altered in the mutant embryonic germ cells (Fig 2C). Compared with the IAP sense and antisense small RNAs and the LINE-1 sense small RNAs (33–40%), LINE-1 antisense small RNAs in the mutant mice were severely decreased (71%) (Fig 2C).

To examine the piRNA expression profile more precisely, we carried out a $^{32}$P labeling assay and NGS analysis of the MILI- and

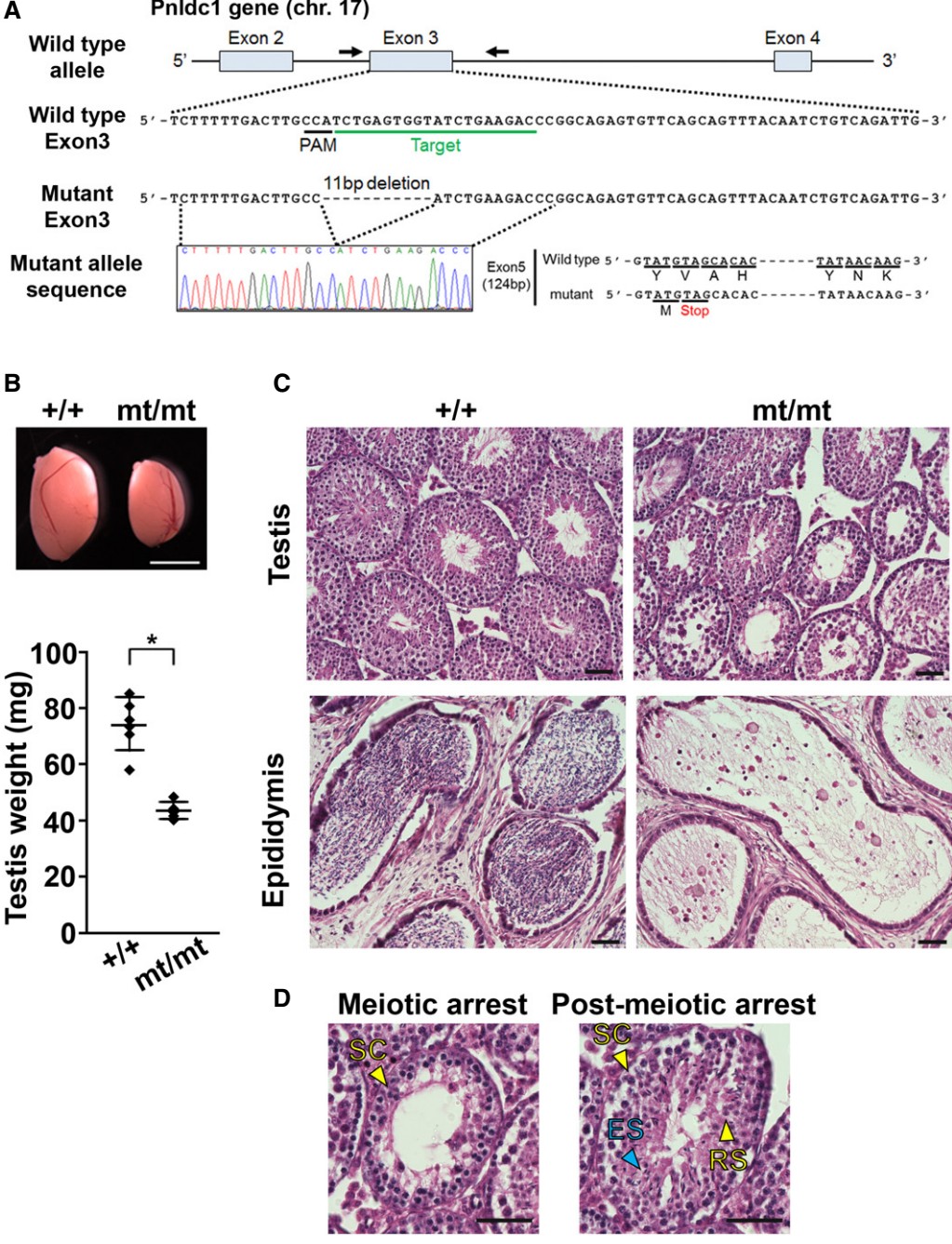

**Figure 1. Generation and phenotypes of *Pnldc1* mutant mice produced by the CRISPR/Cas9 system.**

A  Scheme around exon 3 of mouse *Pnldc1* and its targeted locus. PAM and gRNA-targeted sequences are underlined in black and green, respectively. An 11-nt deletion in the gRNA-targeted region and premature stop codon in *Pnldc1*mt/mt cDNA in exon 5 were confirmed by sequencing.

B  Testicular sizes and weights of adult control and *Pnldc1*mt/mt mice (mean ± SD, *n* = 6, *P* = 0.0003 by *t*-test). Scale bar: 2 mm.

C  Hematoxylin- and eosin-stained sections of adult control and *Pnldc1*mt/mt testes and epididymides. Scale bar: 50 μm.

D  Representative images of meiotic and post-meiotic arrest in *Pnldc1*mt/mt testes. Scale bar: 50 μm. Spermatocyte (SC), round spermatid (RS), and elongating spermatid (ES) were indicated by yellow and blue arrowheads, respectively.

Source data are available online for this figure.

MIWI2-associated small RNAs (Fig EV3). By radiolabeling, we confirmed that both MILI- and MIWI2-bound small RNAs are much longer than the mature piRNA length in the absence of PNLDC1. Moreover, the overall abundance of MIWI2-bound small RNAs was drastically decreased in the PNLDC1-null cells (Fig EV3A). NGS analysis of MILI-bound small RNAs revealed that untrimmed pre-piRNAs were accumulated in both sense and antisense strands in PNLDC1-null cells (Fig EV3B). Moreover, MIWI2-bound small

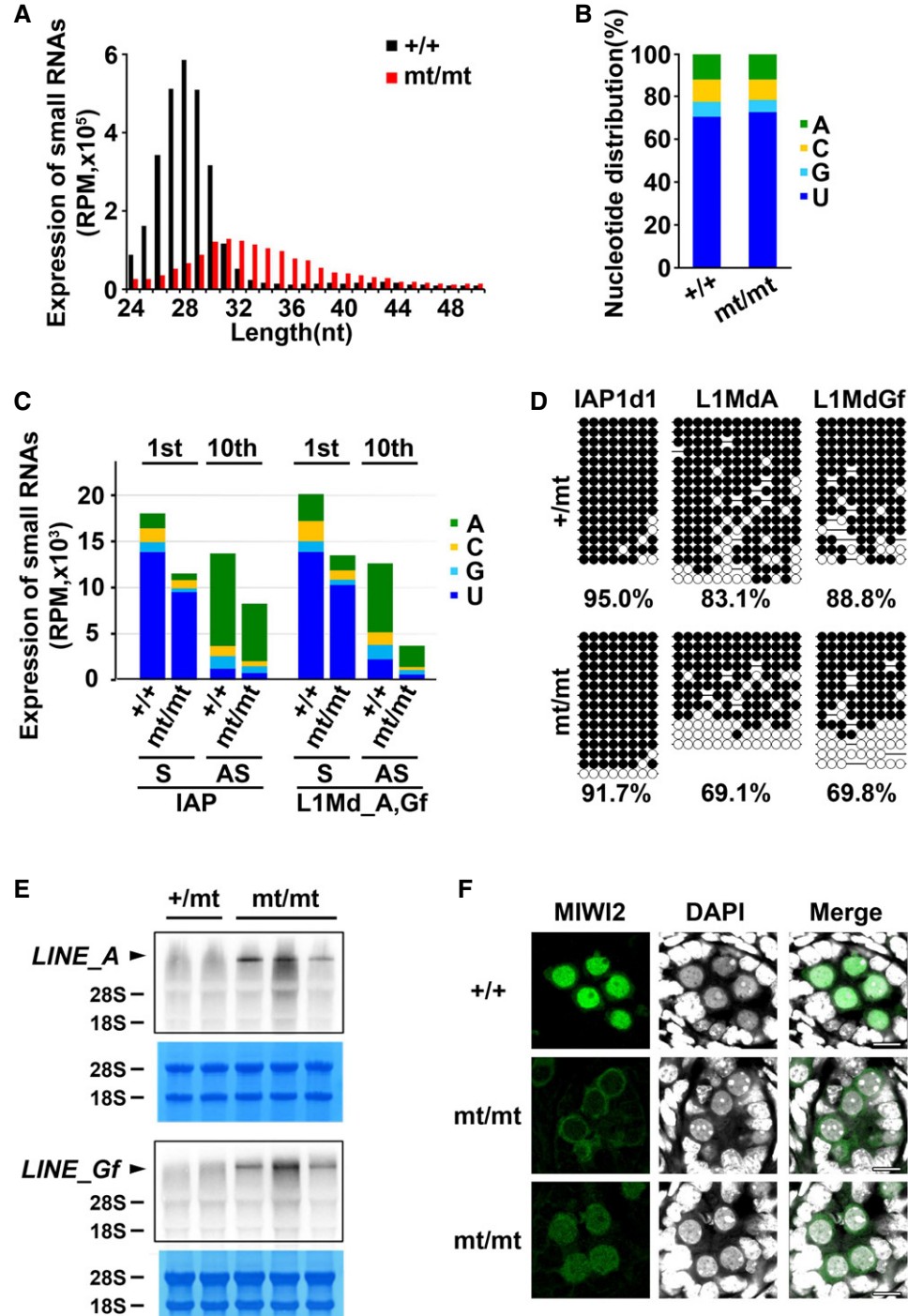

**Figure 2. Small RNAs in embryonic *Pnldc1*^mt/mt testes.**

A   Length distribution of repetitive sequence-derived small RNAs from E16.5 control and *Pnldc1*^mt/mt testes. The small RNAs were analyzed after ribosomal RNA (rRNA) and microRNA (miRNA) mapped reads were removed by piPipes. Black and red bars show the control and *Pnldc1*^mt/mt data, respectively.

B   Nucleotide distribution of the first nucleotide of repetitive sequence-derived piRNAs.

C   Expression and nucleotide distribution of small RNAs corresponding to IAP (M17551) and two LINE-1 retrotransposon (L1Md_A (M13002) and L1Md_Gf (D84391)) sequences.

D   Bisulfite sequencing analysis of the IAP1d1, L1Md_A, and L1Md_Gf genes from purified male germ cells in testes from 10-day-old control and *Pnldc1*^mt/mt mice.

E   Northern blot analysis of L1Md_A and L1Md_Gf transcripts in testes from 14-day-old control and *Pnldc1*^mt/mt mice.

F   Immunostaining of MIWI2 in E16.5 control and *Pnldc1*^mt/mt testes. DNA was stained by DAPI. Scale bar: 10 μm.

Source data are available online for this figure.

RNAs, which should be produced predominantly via the secondary biogenesis, also showed elongated pre-piRNA populations in both strands in the mutant (Fig EV3D). As in the total piRNA pool, there were no significant differences in the distribution of the 1st and 10th nucleotide of MILI- and MIWI2-bound piRNAs, in the presence or absence of PNLDC1 (Fig EV3C and E). These data together suggest that 3′-end trimming by PNLDC1 is required for both primary and secondary piRNA productions. However, it remains unclear why LINE1 antisense piRNAs were so drastically decreased in the total piRNA pool.

## DNA methylation and expression of retrotransposons in *Pnldc1*^mt/mt^ testes

DNA methylation and subsequent gene silencing of the IAP-1Δ1 and LINE-1 retrotransposons are piRNA dependent. Therefore, we examined DNA methylation of the regulatory regions of the IAP and LINE-1 genes in 10-day-old male germ cells, the purity of which was verified by DNA methylation of the H19 gene (Fig EV2B). DNA methylation of IAP genes was not impaired by *Pnldc1* mutation (Fig 2D), and correspondingly, no significant difference in IAP expression was observed (Fig EV2D). Thus, in the case of IAP retrotransposons, it is likely that the remaining antisense piRNAs are sufficient to induce DNA methylation, which was also reported for *Tdrkh*-null and slicer-dead MILI male germ cells [22,25]. In contrast to the IAP genes, DNA methylation of types A and Gf LINE-1 genes was significantly impaired (Fig 2D). This tendency was also observed in *Pnldc1* exon 7 mutant mice (Fig EV2C). Consistent with the reduced DNA methylation, the expression of types A and Gf LINE-1 genes was significantly increased (Figs 2E and EV2D).

piRNA-loaded MIWI2 translocates to the nucleus and induces DNA methylation of retrotransposons in embryonic male germ cells. Therefore, we examined the subcellular localization of MIWI2 in E16.5 male germ cells, in which piRNA-dependent *de novo* DNA methylation occurs. In *Pnldc1*^mt/mt^ male germ cells, nuclear localization of MIWI2 was much weaker compared with that of wild-type germ cells (Fig 2F), despite no effect on the total level of MIWI2 (Fig EV2E). These data are well consistent with the severe reduction in MIWI2-bound piRNAs in the PNLDC1-null cells (Fig EV3A). Taken together, the defective DNA methylation of *Pnldc1*^mt/mt^ male germ cells was due to the reduction in antisense small RNAs corresponding to LINE-1 (Fig 2C) and the decreased nuclear localization of MIWI2 (Fig 2F).

The percentages of DNA methylation of types A and Gf LINE-1 genes in MILI mutant mice were 5–56 and 31–35%, respectively [9,35]. Similarly, those in MitoPLD/Zucchini mutant mice were 16 and 14%, respectively [36,37]. It is notable that the reduction in DNA methylation in *Pnldc1*^mt/mt^ mice was milder than those in MILI and MitoPLD/Zucchini mutant mice, in which piRNA production was markedly reduced. Similarly, *Tdrkh* mutant mice exhibited greater reduction in DNA methylation and higher expression of LINE-1 than those of *Pnldc1*^mt/mt^ mice [22]. This difference would account for the more drastic phenotype of *Tdrkh* mutant mice, in which all male germ cells underwent apoptosis at the pachytene phase. At this point, we cannot rule out the involvement of PARN in piRNA biogenesis in mice. However, it was demonstrated that depletion of PARN has no effect on piRNA maturation in silkworms [23].

## Meiotic arrest of male germ cells caused by *Pnldc1* mutation

Complete loss of piRNA-dependent DNA methylation, which was detected in MILI-, MIWI2-, and TDRKH-null embryonic germ cells, causes total cell death at the pachytene phase [9,12]. However, partial rescue of DNA methylation allows the survival of some cells through the pachytene phase. Indeed, ZF-MIWI2, a fusion protein of MIWI2, and the zinc finger protein recognizing the type A LINE-1 promoter region partly restored the DNA methylation of the type A LINE-1 gene in embryo, and therefore, a fraction of germ cells escaped from apoptosis in MILI-null mice [35].

In *Pnldc1*^mt/mt^ post-natal male germ cells, the DNA methylation level of type A retrotransposons was lower than that in normal mice but was significantly higher than those in MILI, MIWI2, and Tdrkh mutant mice. Considering our ZF-MIWI2 experiment [35], it is conceivable that defective but residual piRNA-dependent DNA methylation allowed approximately half of the male germ cells to escape meiotic arrest. Although the lengths of pre-piRNAs in *Pnldc1* mutant embryonic germ cells varied greatly from 24 to ~45 nt, some of those pre-piRNAs should possess normal piRNA lengths (Fig 2A). It is reasonable to consider that such normal-sized pre-piRNAs in the *Pnldc1* mutant mouse embryos retain normal functions and result in DNA methylation by nuclear localization of MIWI2 (Figs 2D and F, and EV3D).

## Mouse PNLDC1 as the pre-piRNA Trimmer protein of pachytene piRNAs

Next, we analyzed small RNAs 24–50 nt in length in the testes of 24-day-old mice. We focused on pachytene piRNAs, since the vast majority of small RNAs at this stage are pachytene piRNAs (Fig EV4A). Although the distribution of small RNAs in each annotated category was comparable between control and mutant mice (Fig EV4B), there was a striking difference in the size distribution of small RNAs originating from pachytene piRNA clusters (Fig 3A). In control cells, the sizes of the small RNAs were mainly 26–32 nt, which corresponded well to the size of pachytene piRNAs. In contrast, small RNAs in the mutant male germ cells were evenly distributed from 24 to 50 nt with a broad peak (Fig 3A). Because most of these small RNAs are 1U (Fig 3B), the broad size distribution of the small RNAs is presumably due to a defect in 3′-end trimming. These data suggest that pachytene piRNAs do not undergo maturation in the mutant cells. Altogether, PNLDC1 functions as the pre-piRNA Trimmer protein of both embryonic and pachytene piRNAs in male germ cells. Notably, pachytene piRNAs were not detectable in *Tdrkh* mutant mice, in which no germ cells escaped meiotic arrest [22].

MitoPLD/Zucchini catalyzes the endonucleolytic cleavage of piRNA intermediates in PIWI proteins to produce pre-piRNAs [18,19]. Although isolated recombinant MitoPLD/Zucchini is a nonspecific endonuclease [18,19], it is thought that it prefers to cleave before U *in vivo*, generating the 5′ end of a downstream piRNA with 1U bias [21,32]. The 3′ ends of pre-piRNAs are trimmed to the mature length after MitoPLD-mediated cleavage, and hence, pre-piRNAs in *Pnldc1*^mt/mt^ mice are expected to have the 1U bias one nt downstream (+1). To investigate this possibility directly, the nucleotide distribution in the genomic DNA regions up- and downstream of piRNAs in small RNA-sequencing data sets from E16.5 mice was analyzed (Fig EV4C). The small RNAs derived from LTRs and

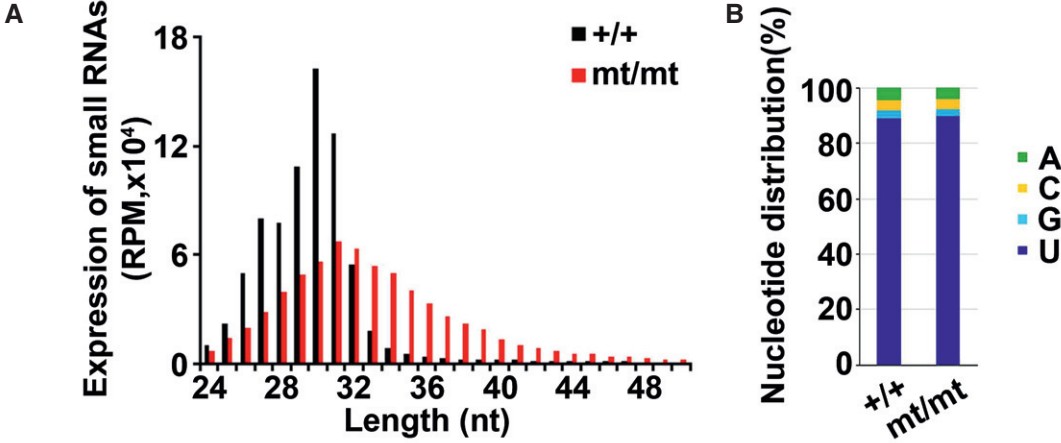

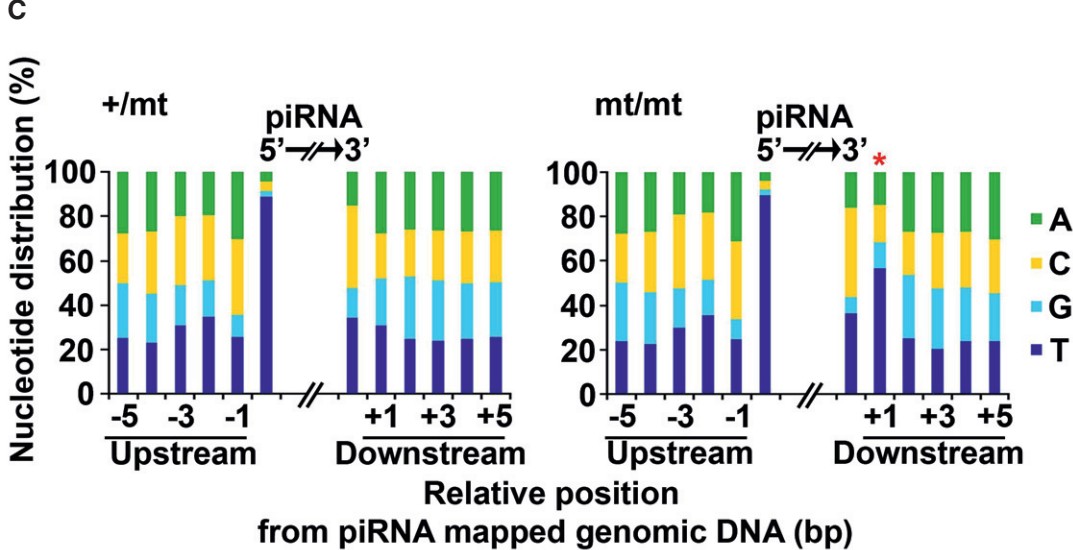

**Figure 3. Small RNAs in adult *Pnldc1*[mt/mt] male germ cells.**

A   Length distribution of small RNAs derived from pachytene piRNA clusters from the testes of 24-day-old control and *Pnldc1*[mt/mt] mice. The small RNAs were analyzed after rRNA and miRNA mapped reads were removed by piPipes. Black and red bars show the control and *Pnldc1*[mt/mt] data, respectively.

B   Nucleotide distribution of the first nucleotide of the small RNAs.

C   Nucleotide distribution around the small RNAs. Asterisk (*) indicates strong T bias at the +1 position.

LINE-1 in *Pnldc1*[mt/mt] cells displayed a strong T bias at the +1 position, while no such bias was observed in the control. This phenotype was also detected in *Tdrkh* mutant mice [21,32]. Importantly, the same +1T bias was observed in the small RNAs of adult *Pnldc1*[mt/mt] mice (Fig 3C), indicating that similar to embryonic piRNAs, pachytene piRNAs also undergo initial cleavage by MitoPLD/Zucchini and then maturation by PNLDC1-dependent 3′-end trimming.

**Post-meiotic arrest of adult male germ cells due to *Pnldc1* mutation**

All male germ cells that survived through the meiotic phase underwent post-meiotic arrest in the *Pnldc1* mutant. To explore the reason for this, we examined the expression of MILI and MIWI proteins in mutant adult testes by Western blotting and immunoprecipitation analyses. Although the MIWI protein was detectable after immuno-precipitation, its level was strikingly lower in the mutant testes than in the control (Fig 4A). Because no significant differences in the MIWI and MILI mRNA levels were found (Fig 4B), this decrease was presumed to occur at the post-transcriptional level.

We next analyzed the MILI- and MIWI-bound piRNAs in the testes of 24-day-old mice by immunoprecipitation and end-radio-labeling (Fig 4C). As expected,, the lengths of MILI-bound piRNAs were longer in the *Pnldc1*[mt/mt] testes than those in the control, reflecting the defective trimming activity. We found that the 3′ end of those untrimmed MILI-bound RNAs in the PNLDC1 mutant was

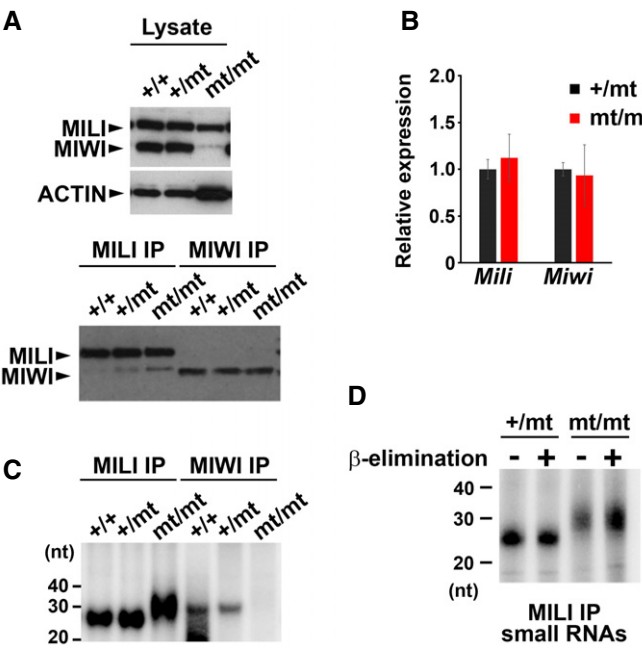

**Figure 4.  Expression and small RNA binding of mouse PIWI-family proteins in *Pnldc1*^mt/mt mice.**

A, B   Expression analyses of MILI and MIWI in 24-day-old testes by Western blotting (A) and RT–qPCR (B). (A) β-ACTIN was for the loading control. (B) Relative expression was normalized to that of *β-Actin*. Bars show mean ± SEM (*n* = 3). *P* = 0.64 for *Mili*, *P* = 0.83 for *Miwi* by *t*-test.

C   MILI- and MIWI-bound small RNAs in testes from 24-day-old control and *Pnldc1*^mt/m mice. The RNAs that co-precipitated with MILI and MIWI were purified and separated by 15% denatured acrylamide gel electrophoresis after ^32P-end-labeling.

D   β-elimination assay. ^32P-end-labeled MILI-bound piRNAs from control and *Pnldc1*^mt/mt 24-day-old testes were separated by 12% denatured acrylamide gels with or without β-elimination treatment.

Source data are available online for this figure.

2′-*O*-methylated (Fig 4D), just like untrimmed RNAs upon Trimmer or BmPapi depletion in silkworms [23]. It has been shown that, in silkworms, the piRNA methyltransferase Hen1 can only methylate relatively short pre-piRNA populations, close to the mature piRNA length; pre-piRNA populations that are too long to be methylated are prone to degradation and thus do not survive, whereas relatively short pre-piRNA populations that can be marginally methylated are stable and thus become detectable in the piRNA pool. We conclude that this is also likely the case in mice.

Pachytene piRNAs are normally loaded onto both MILI and MIWI [38]. Unexpectedly, however, no MIWI-bound piRNAs or pre-piRNAs were detected in adult *Pnldc1*^mt/mt male germ cells (Fig 4C). One possible explanation for the absence of MIWI-bound RNAs is that pre-piRNAs of longer lengths cannot be stably loaded onto MIWI because of the lack of trimming activity. The other, and not mutually exclusive, possibility is enhanced degradation of MIWI. It was reported that MIWI protein expression is regulated by ubiquitin-/proteasome-mediated degradation in late spermatids, and this degradation is triggered by loading of piRNA onto MIWI [39]. Because the detected MIWI did not associate with piRNAs in the *Pnldc1* mutant (Fig 4C), only unloaded MIWI might have remained. In either case, we speculate that the severe reduction in MIWI

and/or the loss of MIWI-bound piRNAs led to the post-meiotic arrest in *Pnldc1*^mt/mt mice.

In summary, we showed that PNLDC1 plays an essential role throughout the development of mouse male germ cells by acting as the pre-piRNA Trimmer protein. Our data cast new insights into the molecular and biological functions of piRNA maturation in spermatogenesis.

## Note

Two papers studying the function of PNLDC1 have been published during the revision of this paper [40,41]. Their conclusions are essentially the same as those in our current study; namely, PNLDC1-null mice showed the defects of piRNA 3′ end trimming, transposon gene silencing, and spermatogenesis.

## Materials and Methods

### Mice

Mutant mice were generated using the CRISPR/Cas9 system. Cas9 mRNA and each single-guide RNA (sgRNA) transcribed *in vitro* by the mMESSAGE mMACHINE T7 Ultra Kit (Ambion) were injected into C57BL6 zygotes. Two-cell-stage embryos were transplanted into pseudopregnant mice. The sgRNA-targeting sequences and PCR primers are listed in Table EV1. All animal experiments were performed in accordance with the general guidelines of The Institute of Experimental Animal Sciences, Osaka University Medical School.

### Antibodies

Anti-MIWI2 primary antibody (ab21869, Abcam) and goat anti-rabbit Alexa fluor 488 secondary antibody (A11008, Thermo Fisher Scientific) were used for immunofluorescence staining. Anti-MILI antibody (PM044, MBL) and anti-MIWI antibody (2C12, Wako) were used for immunoprecipitation. The affinity-purified anti-Mili26F antibody recognizing both MILI and MIWI [14] was used for Western blotting. PE anti-mouse CD326 (Ep-CAM) (G8.8, BioLegend) was used for germ cell sorting.

### Next-generation sequencing analysis

Small RNA-sequencing by HiSeq from E16.5 and post-natal day 24 testes were described elsewhere [35]. For paired-end libraries, only perfectly complementary paired reads were extracted for analysis. piPipes was utilized for small RNA-sequencing analysis [32]. The parameters used were as follows: zero mismatches allowed for ribosomal RNA mapping, hairpin mapping, and genome mapping; and up to two mismatches allowed for repeat mapping. Mm9 was used as the mouse genome reference sequence. The lengths of microRNAs (miRNAs) and small RNAs were defined as 20–24 and 24–50 nt, respectively, and a summary is shown in Table EV4. Bowtie was used with the default settings to map representative LINE-1 retrotransposons (L1Md_A; M13002 and L1Md_Gf; D84391) and IAP (M17551) [42]. For MILI- and MIWI2-bound small RNA analyses, RNAs were purified from immunoprecipitated complexes and subjected to small RNA-seq libraries preparation with SMARer

smRNA-Seq Kit (Clontech). The single-read (SR) 51-nucleotide (nt) raw sequence data were processed by standard Illumina pipelines. The SR reads were mapped to mm9 or TE consensus sequence by Bowtie after adapter trimming by cutadapt. The aligned reads were analyzed by SeqMonk and Excel.

### Bisulfite sequencing

Ep-CAM-positive germ cells from testes of 10-day-old mice were sorted by BD FACSAria™ IIu (BD Biosciences), and bisulfite analysis was performed as described previously [35]. PCR primer sequences are listed in Table EV2.

### Northern blotting

Total RNA was purified from whole testes of 14-day-old mice by Isogen (Nippon Gene). After crosslinking, filter was stained with methylene blue for visualized 28S and 18S ribosomal RNAs. Northern blotting was performed according to a previously published method [9].

### Reverse-transcription quantitative PCR (RT-qPCR)

cDNAs were obtained by reverse transcription of purified RNA samples using the ThermoScript™ RT-PCR system (Thermo Fisher Scientific). RT–qPCR was performed using the THUNDERBIRD™ SYBR qPCR Mix (TOYOBO) with the appropriate primers (Table EV3) and the CFX384 Touch™ Real-Time PCR Detection System (Bio-Rad).

### Western blotting

Testes from E16.5 control and *Pnldc1*$^{mt/mt}$ mice were homogenized by syringe in RIPA buffer (50 mM Tris pH 7.5, 150 mM NaCl, 1% NP40, 0.1% SDS, 0.25% sodium deoxycholate, 1 mM EDTA) and treated by Bezonase (Merck). Then, 0.5 testis lysate per lane was subjected to 5–20% SDS–PAGE. Anti-MIWI2 (Miwi2-C) and anti-β-actin (A5441, Sigma) antibodies were used as the primary antibodies for Western blotting.

### Histochemistry and immunofluorescence

Testes were fixed in Bouin's solution (Sigma) before embedding in paraffin for hematoxylin and eosin staining. The image was obtained using the BZ-X710 microscope (Keyence). The numbers of cells presenting with meiotic arrest or post-meiotic arrest were counted using ImageJ. For immunofluorescence, testes were fixed in 2% PFA/PBS for 1 h before embedding in OCT compound. Before blocking, permeabilization in 0.5% Triton X-100/PBS for 30 min at room temperature and antigen retrieval of the sections in 1.8 mM $C_6H_8O_7 \cdot H_2O$, 8.2 mM $C_6H_5Na_3O_7 \cdot 2H_2O$ by autoclaving for 20 min at 121°C were performed. Cryosections were treated as described previously [35].

### Western blotting, immunoprecipitation, and detection of small RNAs

Immunoprecipitation and detection of small RNAs from the testes of 24-day-old mice using anti-MILI and anti-MIWI antibodies were performed as described previously [9]. Anti-Mili26F antibody was used to detect MILI and MIWI by Western blotting.

### Analysis of RNA 3′ ends

$NaIO_4$ reaction was performed basically as described previously [24]. Anti-MILI antibody precipitated small RNAs from adult testis were suspended in 20 μl of borax/boric acid buffer (pH 8.6) containing 25 mM $NaIO_4$ at room temperature for 30 min. 2 μl of glycerol was added to quench unreacted $NaIO_4$. β-elimination was then performed by adding 2 μl of 500 mM NaOH followed by incubation at 45°C for 90 min. The resultant RNA was collected by ethanol precipitation. RNAs were labeled by γ-$^{32}$P-ATP and run on 8 M urea-containing 12% denatured acrylamide gel.

### Data availability

The E16.5 and post-natal day 24 small RNA data sets and MILI- and MIWI2-associated small RNA data sets from this study have been deposited in DDBJ under accession number DRA005930 and DRA006399.

**Expanded View** for this article is available online.

## Acknowledgements

The authors thank Ms. N. Asada for her technical assistance and Ms. M. Imaizumi for her secretarial work. This work was supported in part by a Grant-in-Aid for Scientific Research on Research (A) (#15H02509), Research (B) (#15H04699) from MEXT/JSPS, and Core Research for Evolutional Science and Technology (CREST) (#J150701424) from AMED, Japan, and Grant-in-Aid for Scientific Research on Innovative Areas ('Non-coding RNA neo-taxonomy') 26113007 to Y. T. and Grant-in-Aid Young Scientists (B) 17K17673 to N. I. This work was also supported by Center for Medical Research and Education (CentMeRE), Graduated school of Medicine, Osaka University.

## Author contributions

YT, NI, SK-M, and TNakan conceived the study, and TNi, IN, and SK-M designed and performed experiments. TNi and TNakat generated Pnldc1 mutant mice. IN carried out NGS data analysis. All authors analyzed data. TNi, IN, SK-M, and TNakan wrote the manuscript with feedback from all co-authors.

## Conflict of interest

The authors declare that they have no conflict of interest.

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
