## [Review Process File · EMBO Reports]

PNLDC1, Mouse pre-piRNA Trimmer, is Required for Meiotic and Post-meiotic Male Germ Cell Development

Toru Nishimura, Ippei Nagamori, Tsunetoshi Nakatani, Natsuko Izumi, Yukihide Tomari, Satomi Kuramochi-Miyagawa, Toru Nakano

Review timeline:

Submission date:	4 August 2017
Editorial Decision:	8 September 2017
Revision received:	8 December 2017
Editorial Decision:	21 December 2017
Revision received:	5 January 2018
Accepted:	17 January 2018

Transaction Report:

1st Editorial Decision

8 September 2017

Thank you for the submission of your research manuscript to our journal. We have now received the full set of referee reports that is copied below.

As you will see, the referees acknowledge the potential interest of the findings. However, referees 1 and 3 also point out several technical concerns and have a number of suggestions for how the study should be strengthened, and I think that all of them should be addressed.

Given these constructive comments, we would like to invite you to revise your manuscript with the understanding that the referee concerns (as detailed above and in their reports) must be fully addressed and their suggestions taken on board. Please address all referee concerns in a complete point-by-point response. Acceptance of the manuscript will depend on a positive outcome of a second round of review. It is EMBO reports policy to allow a single round of revision only and acceptance or rejection of the manuscript will therefore depend on the completeness of your responses included in the next, final version of the manuscript.

Revised manuscripts should be submitted within three months of a request for revision; they will otherwise be treated as new submissions. Please contact us if a 3-months time frame is not sufficient for the revisions so that we can discuss the revisions further.

Supplementary/additional data: The Expanded View format, which will be displayed in the main HTML of the paper in a collapsible format, has replaced the Supplementary information. You can submit up to 5 images as Expanded View. Please follow the nomenclature Figure EV1, Figure EV2 etc. The figure legend for these should be included in the main manuscript document file in a section called Expanded View Figure Legends after the main Figure Legends section. Additional Supplementary material should be supplied as a single pdf labeled Appendix. The Appendix

includes a table of content on the first page, all figures and their legends. Please follow the nomenclature Appendix Figure Sx throughout the text and also label the figures according to this nomenclature. For more details please refer to our guide to authors.

Regarding data quantification, can you please specify the number "n" for how many experiments were performed, the bars and error bars (e.g. SEM, SD) and the test used to calculate p-values in the respective figure legends? This information is currently incomplete and must be provided in the figure legends. Please also include scale bars in all microscopy images.

We now strongly encourage the publication of original source data with the aim of making primary data more accessible and transparent to the reader. The source data will be published in a separate source data file online along with the accepted manuscript and will be linked to the relevant figure. If you would like to use this opportunity, please submit the source data (for example scans of entire gels or blots, data points of graphs in an excel sheet, additional images, etc.) of your key experiments together with the revised manuscript. Please include size markers for scans of entire gels, label the scans with figure and panel number, and send one PDF file per figure or per figure panel.

As part of the EMBO publication's Transparent Editorial Process, EMBO reports publishes online a Review Process File to accompany accepted manuscripts. This File will be published in conjunction with your paper and will include the referee reports, your point-by-point response and all pertinent correspondence relating to the manuscript.

I look forward to seeing a revised version of your manuscript when it is ready. Please let me know if you have questions or comments regarding the revision.

REFeree REPORTS

Referee #1:

The manuscript by Nishimura et al. centers on the function of mouse PNLDC1, an orthologue of the *Bombyx mori* piRNA trimmer. The authors generate a *PnlDC1* null allele and analyze the consequences of *PnlDC1*-deficiency for embryonic and pachytene piRNA pathways. The loss of PNLDC1 results in defective spermatogenesis with half the tubules showing a meiotic arrest and the other half resulting in spermiogenic arrest. The authors present a second CRISPR-null allele to confirm this result. Loss of PNLDC1 results in hypomorphic embryonic and pachytene piRNA pathways. The findings are potentially important but the current manuscript lacks sufficient experimentation/depth to accurately define the consequences of PNLDC-deficiency to piRNA biogenesis pathways; thus, I cannot fully endorse the manuscript for publication at this point.

Concerns

1. Statement: 'This tendency was also observed in *PnlDC1* exon 7 mutant mice (Supplemental Fig. 2D). Consistent with the reduced DNA methylation, the expression of types A and Gf LINE-1 genes was significantly increased (Fig. 2E)'. The authors only use qRT-PCR to support this result. Northern blotting should be used to confirm if full length L1 transcripts are observed that are capable of transposition. Likewise, anti-L1 Orf1 antibody should be used for IF to confirm potential defective L1 silencing at E16.5.

2. Statement: 'These results suggest that the defective DNA methylation of *PnlDC1*mt/mt male germ cells was due to the reduction of antisense small RNAs corresponding to LINE-1 (Fig. 2C) and the decreased nuclear localization of MIWI2 (Fig. 2F)'. The authors did not directly measure piRNAs within MIWI2 RNPs, so the above conclusion is indirectly inferred. I think it would be important to actually sequence MILI and MIWI2-bound piRNAs to directly and accurately assess the impact of

PNLDC-deficiency on embryonic piRNA biogenesis/amplification. I think this is a key point without which the manuscript lacks depth. I also think the piRNAs bound to MILI and MIWI RNPs should be 32P labelled and visualized to give an estimation of the reduction in piRNA loading in the respective RNPs. Lastly, piRNA loading in MIWI2 permits nuclear localization; I don't think MIWI2 distinguishes if they are antisense or not.

3. Both MILI and MIWI slice L1 transcripts in spermatocytes and round spermatids, respectively (DiGiacomo et al, 2013 and Reuter et al, 2011). Is L1 deregulated in PNLDC1-deficient spermatocytes and round spermatids? Anti-L1 Orf1 antibody should be used for IF to confirm defective L1 silencing in PNLDC1-deficient spermatocytes and round spermatids. Northern blotting should be used to confirm if full length L1 transcripts are expressed in adult PNLDC1-deficient testis.

Minor concerns

1. Statement: 'MILI is constitutively expressed in embryonic testes from the primordial germ cell to round spermatid stage (Kuramochi-Miyagawa et al. 2001)'. This statement is not fully precise, MILI expression is absent in leptotene and zygotene stages of meiotic prophase (DiGiacomo et al, 2013).

2. Statement: 'Thus, in the case of IAP retrotransposons, it is likely that the remaining antisense piRNAs are sufficient to induce DNA methylation, which was also reported for Tdrkh-null male germ cells (Saxe et al. 2013)'. Same is true for slicer-dead MILI mice, amplification is not required to silence IAPs.

Referee #2:

In their manuscript titled "PNLDC1, Mouse pre-piRNA Trimmer, is Required for Meiotic and Post-meiotic Male Germ Cell Development" Nishimura et al. investigate the function of mouse PNLDC1, a paralog of the poly(A) specific ribonuclease (PARN), in piRNA biogenesis during spermatogenesis. Nishimura and colleagues generated and characterized multiple PNLDC1 mutant alleles using CRIPSR/Cas genome editing. PNLDC1 Mutants are viable but male sterile and exhibit defects in spermatogenesis analogous to Mili and Miwi mutants. PNLDC1 mutant germ cells exhibit reduced DNA methylation at LINE1 elements and loss of restriction of these mobile genetic elements. Nishimura and colleagues directly investigate piRNAs and their PIWI protein partners in PNLDC1 mutant testes, and observe reduced mature PIWI-piRNA complexes and longer piRNAs. Interestingly, meiotic and post-meiotic defects were observed simultaneously suggesting that trimming of piRNAs is required for efficiency of piRNA silencing, but that untrimmed piRNA-PIWI complexes can partially support transposon silencing. This work clearly establishes mmPNLDC1 as pre-piRNA trimmer for pre-pachytene and pachytene piRNAs, and elegantly characterizes its requirement for stable PIWI-piRNA complex formation and piRNA-mediated transposon restriction in mammals. This concise work adds important information to our understanding of germ cell biology in general and mechanisms of piRNA-mediated transposon control in particular. I strongly support the publication of this manuscript without major changes.

Minor points:

1. Fig.4A would benefit from a loading control (e.g. anti-Tubulin or anti-Actin WB)

2. The second sentence in the abstract could be slightly re-phrased to clarify that pre-piRNAs are only loaded into PIWI proteins after MitoPLD/Zuc cleavage. Suggestion: "In piRNA biogenesis, the endonuclease MitoPLD/Zucchini cleaves long, single-stranded RNAs to generate 5' termini of precursor piRNAs (pre-piRNAs) that are consecutively loaded into PIWI-family proteins. "

Referee #3:

Nishimura et al describe the defects observed in pnd1 mutant mice. Based on previous work, PNLDC1, together with PARN, is a strong candidate for piRNA 3' trimming. The authors make a strong case for PNLDC1 as the trimming enzyme in mice. Nevertheless, a role for PARN cannot be ruled out.

The developmental and molecular phenotypes that are described are what can be expected. There are no big surprises. All observations match what can be expected for a factor that trims piRNAs at their

3' end. Both embryonic and pachytene piRNA populations are described, basically with the same outcome.

Overall, the manuscript is clear and to-the-point. I do not see the need for major changes, but I do hope the authors can address the below points, in order to make the manuscript more concise and complete.

-Page 9-10:

Why would IAP anti-sense piRNAs be more strongly affected than LINE1 anti-sense piRNAs? The explanation that is given for the decrease in anti-sense LINE1 piRNAs (decreased cleavage activity by MILI) holds just as well for IAP. I understand that it will be hard to explain the difference, but the current text provides an apparent explanation that does not seem to be the right/complete one. Please rephrase to prevent any mis-understandings.

Related to this issue: the stronger decrease of anti-sense piRNAs compared to sense (also for IAP!) likely also is related to a simple direct effect of PNL1 on secondary piRNAs. Most of these likely also need 3' trimming. This aspect is not mentioned, but may be equally important (perhaps even more important) than a potential decrease in slicing activity of MILI. This might be addressed even through bioinformatics analysis. Are the MIWI2 piRNAs that have a downstream MILI site within 30 nucleotides less affected by PNL1 than those that do not? Reasoning being that such a close-by second downstream MILI site would release a piRNA that does not need 3' trimming.

-A role for PARN in piRNA trimming cannot be ruled out. Given the relatively weak phenotype of the mice, it is well possible that PARN can participate in trimming as well. This should be discussed briefly.

-Are the 3' ends of the piRNA intermediates methylated? On page 9 the authors suggest they are not, based on analogy to another system. However, this is not addressed directly (through experiment) in the mouse. Ideally, the authors show the 2'O methylation status of, for example, the MILI-bound intermediates in the mutant, or they must make clear that this remains unresolved.

-Please label histology figures for non-mouse researchers. What does one see where?

-On page 13 the authors write: "To determine if the post-meiotic phenotype is caused by impaired pachytene piRNA production...". This question remains unresolved. What follows are experiments showing that there are pachytene piRNA defects, but there are no experiments addressing the causality. Please rephrase, to make the text more accurate.

1st Revision - authors' response

8 December 2017

Comments of the referees and our responses

Referee #1:

1. Statement: 'This tendency was also observed in Pnl1 exon 7 mutant mice (Supplemental Fig. 2D). Consistent with the reduced DNA methylation, the expression of types A and GfLINE-1 genes was significantly increased (Fig. 2E)'. The authors only use qRT-PCR to support this result. Northern blotting should be used to confirm if full length L1 transcripts are observed that are capable of transposition. Likewise, anti-L1 Orf1 antibody should be used for IF to confirm potential defective L1 silencing at E16.5.

Following the referee's comment, we confirmed the upregulation of L1 genes by Northern blotting and substituted the Fig 2E by the image of Northern blotting. The data of qRT-PCR was moved to the Fig EV2E. We rewrote the corresponding parts in the Figure Legend (p28 line 5) and the Materials and Methods (p19 line 8-12). We also carried out IF staining with a commercially available anti-ORF2 antibody (Santacruz #sc-67198). However, unfortunately, expression of L1 protein was not detectable even in the MIWI2-deficient E16.5 testis (data not shown).

2. Statement: 'These results suggest that the defective DNA methylation of *Pnlcd1mt/mt* male germ cells was due to the reduction of antisense small RNAs corresponding to *LINE-1* (Fig. 2C) and the decreased nuclear localization of *MIWI2* (Fig. 2F)'. The authors did not directly measure piRNAs within *MIWI2* RNPs, so the above conclusion is indirectly inferred. I think it would be important to actually sequence *MILI* and *MIWI2*-bound piRNAs to directly and accurately assess the impact of *PNLDC*-deficiency on embryonic piRNA biogenesis/amplification. I think this is a key point without which the manuscript lacks depth. I also think the piRNAs bound to *MILI* and *MIWI* RNPs should be ³²P labelled and visualized to give an estimation of the reduction in piRNA loading in the respective RNPs. Lastly, piRNA loading in *MIWI2* permits nuclear localization; I don't think *MIWI2* distinguishes if they are antisense or not.

We appreciate the referee's very constructive comments and carried out the ³²P labelling assay and NGS analysis (Fig EV3). ³²P labelling assay clearly demonstrated that both *MILI*- and *MIWI2*-bound small RNAs are much longer than the mature piRNA length in the absence of *PNLDC1*. Moreover, we observed a severe reduction of *MIWI2*-bound piRNAs in the *PNLDC1*-null cells (Fig EV3A), which is well consistent with the impaired nuclear localization of *MIWI2* in the mutant mice. In addition, our NGS analysis verified the referee's assumption that *MIWI2* does not distinguish sense and antisense as shown in Fig EV3E and F. We added the sentences regarding these points (p9 line 5-p10 line 2, p11 line 5-7) and relevant descriptions in the Materials and Methods (p18 line 14-p19 line 3) and in the EV Figure Legends (p31 line 11-p32 line 3).

3. Both *MILI* and *MIWI* slice *L1* transcripts in spermatocytes and round spermatids, respectively (DiGiacomo et al, 2013 and Reuter et al, 2011). Is *L1* deregulated in *PNLDC1*-deficient spermatocytes and round spermatids? Anti-*L1* Orf1 antibody should be used for IF to confirm defective *L1* silencing in *PNLDC1*-deficient spermatocytes and round spermatids. Northern blotting should be used to confirm if full length *L1* transcripts are expressed in adult *PNLDC1*-deficient testis.

As written in the response to the comments of #1, although we were not able to detect the *L1* protein by the commercially available antibody, we confirmed the upregulation of *L1* genes by Northern blotting, agreeing well with our original qRT-PCR data. We appreciate if the referee understands the situation.

Minor concerns

1. Statement: '*MILI* is constitutively expressed in embryonic testes from the primordial germ cell to round spermatid stage (Kuramochi-Miyagawa et al. 2001)'. This statement is not fully precise, *MILI* expression is absent in leptotene and zygotene stages of meiotic prophase (DiGiacomo et al, 2013).

We would like to thank for the kind suggestion. We changed the corresponding sentence (p3 line 9).

2. Statement: 'Thus, in the case of *IAP* retrotransposons, it is likely that the remaining antisense piRNAs are sufficient to induce DNA methylation, which was also reported for *Tdrkh*-null male germ cells (Saxe et al. 2013)'. Same is true for slicer-dead *MILI* mice, amplification is not required to silence *IAPs*.

Thank you for your helpful comment. This point is added in the corresponding sentence (p10 line 12).

Referee #2:

Minor points:

#1. Fig. 4A would benefit from a loading control (e.g. anti-Tubulin or anti-Actin WB)

Following the comment, we added the data of beta-actin as a control of the Western blotting data analysis (Fig 4A) and added a sentence on the Figure Legend (p29 line 3).

#2. The second sentence in the abstract could be slightly re-phrased to clarify that pre-piRNAs are only loaded into PIWI proteins after MitoPLD/Zuc cleavage. Suggestion: "In piRNA biogenesis, the endonuclease MitoPLD/Zucchini cleaves long, single-stranded RNAs to generate 5' termini of precursor piRNAs (pre-piRNAs) that are consecutively loaded into PIWI-family proteins. "

We appreciate your helpful suggestion and have replaced the sentence (p2 line 4-6).

Referee #3:

1, -Page 9-10:

Why would IAP anti-sense piRNAs be more strongly affected than LINE1 anti-sense piRNAs? The explanation that is given for the decrease in anti-sense LINE1 piRNAs (decreased cleavage activity by MILI) holds just as well for IAP. I understand that it will be hard to explain the difference, but the current text provides an apparent explanation that does not seem to be the right/complete one. Please rephrase to prevent any mis-understandings.

We assume that the referee intended to point out the stronger reduction of LINE1 anti-sense piRNAs compared to IAP anti-sense piRNAs. We fully agree that defects in secondary piRNA biogenesis holds true not only for LINE1 anti-sense piRNAs but also for IAP anti-sense piRNAs and that in itself cannot explain the difference between them. At this point, it remains unclear why LINE1 anti-sense piRNAs were more severely decreased than IAP anti-sense piRNAs in the total piRNA pool. Therefore, to avoid mis-understandings, we deleted the sentence "Considering that antisense piRNAs with a 10A bias are produced by secondary biogenesis, the severe decrease in LINE-1 antisense RNAs may be attributed to impaired secondary piRNA biogenesis (Supplemental Fig. S2B)" from our manuscript. We are sorry for the confusion and appreciate the referee's valuable comment. We have added a sentences in the Results and Discussion (p9 line 18-p10 line 2).

2, -Related to this issue: the stronger decrease of anti-sense piRNAs compared to sense (also for IAP!) likely also is related to a simple direct effect of PNLDC1 on secondary piRNAs. Most of these likely also need 3' trimming. This aspect is not mentioned, but may be equally important (perhaps even more important) than a potential decrease in slicing activity of MILI. This might be addressed even through bioinformatics analysis. Are the MIWI2 piRNAs that have a downstream MILI site within 30 nucleotides less affected by PNLDC1 than those that do not? Reasoning being that such a close-by second downstream MILI site would release a piRNA that does not need 3' trimming.

We thank the referee for the valuable suggestion. Indeed, our new NGS analysis of MILI-bound small RNAs showed that untrimmed pre-piRNAs were accumulated in both sense and anti-sense strands in PNLDC1-null cells (Fig. EV3D). Moreover, MIWI2-bound small RNAs, which should be produced predominantly via the secondary biogenesis, also showed elongated pre-piRNA populations in PNLDC1-null cells (Fig. EV3). These data suggest that, as the referee pointed out, the trimming activity of PNLDC1 is directly required also for secondary piRNA production. We have added a paragraph regarding this point in the revised manuscript (p9 line 10-18).

3, -A role for PARN in piRNA trimming cannot be ruled out. Given the relatively weak phenotype of the mice, it is well possible that PARN can participate in trimming as well. This should be discussed briefly.

We agree with the referee that, at this point, the involvement of PARN in piRNA trimming cannot be ruled out in mice. However, it was demonstrated that depletion of PARN has no effect on piRNA maturation in silkworms (Izumi et al., 2016, Fig. S1D), and we assume that the situation is similar in mice. We have added a brief description of this point in the revised manuscript. (p11 line18- p12 line 2).

4, -Are the 3' ends of the piRNA intermediates methylated? On page 9 the authors suggest they are not, based on analogy to another system. However, this is not addressed directly (through experiment) in the mouse. Ideally, the authors show the 2'O methylation status of, for example, the MILI-bound intermediates in the mutant, or they must make clear that this remains unresolved.

To examine the methylation status, we performed b-elimination assay of the MILI associated small RNAs in the adult testes. First, we verified the feasibility of the method in our hands using synthesized oligo RNA (Fig EV4D). As shown in Fig 4E, MILI-bound untrimmed pre-piRNAs in the PNLDC1 mutant were confirmed to be 2'-O-methylated, just like untrimmed RNAs upon Trimmer or BmPapi depletion in silkworms (Izumi et al. 2016). It was previously shown that, in silkworms, the piRNA methyltransferase Hen1 can only methylate relatively short pre-piRNA populations, close to the mature piRNA length; longer pre-piRNA populations that cannot be unmethylated are prone to degradation and thus do not survive, whereas relatively short pre-piRNA populations that can be (marginally) methylated are stable and become detectable in the piRNA pool. We believe this is also the case in mice. We have added several sentences in the Results and Discussion (p15 line 12-p16 line 2), the Materials and Methods (p21 line 8-15), the Figure Legends (p29 line 7-10) and in the EV Figure Legends (p32 line 14-16). And we have rewrote several parts in the following paragraph (p16 line 3-14).

5, -Please label histology figures for non-mouse researchers. What does one see where?

We labeled SC (spermatocyte) as meiotic germ cell, and RS (round spermatid) and ES (elongated spermatid) as post meiotic germ cells in Fig 1D. We added a sentence in Figure Legends (p27 line 11-12).

6, -On page 13 the authors write: "To determine if the post-meiotic phenotype is caused by impaired pachytene piRNA production...". This question remains unresolved. What follows are experiments showing that there are pachytene piRNA defects, but there are no experiments addressing the causality. Please rephrase, to make the text more accurate.

We apologize for the confusion. As the referee pointed out, there is no clear evidence showing the relationship between the post-meiotic phenotype and the impaired pachytene piRNA production. We deleted the phrase "To determine if the post-meiotic phenotype is caused by impaired pachytene piRNA production," together with the former sentence "As shown in Fig 1C, approximately half of the male germ cells in *PnlDC1^{mt/mt}* mice underwent meiosis but were arrested at the post-meiosis stage." from the original manuscript.

2nd Editorial Decision

21 December 2017

Thank you for the submission of your revised manuscript to EMBO reports. Your manuscript has been sent back to former referees 1 and 3; please find their reports copied below.

As you will see both referees are very positive about the study and support publication in EMBO reports without further revision.

Browsing through the manuscript myself, I noticed a few minor editorial changes that we need before we can proceed with the official acceptance of your study.

We look forward to seeing a final version of your manuscript as soon as possible. Please let me know if you have questions or comments regarding the revision.

REFeree REPORTS

Referee #1:

All my concerns have been addressed

Referee #3:

The authors have addressed the indicated issues adequately. I support publication in EMBO rep.

2nd Revision - authors' response

5 January 2017

The authors addressed the minor editorial changes.

Corresponding Author Name: Toru Nakano, Satomi Kuramochi-Miyagawa

Manuscript Number: EMBOR-2017-44957v3